# Whole Genome Sequencing Contributions and Challenges in Disease Reduction Focused on Malaria

**DOI:** 10.3390/biology11040587

**Published:** 2022-04-13

**Authors:** Olusegun Philip Akoniyon, Taiye Samson Adewumi, Leah Maharaj, Olukunle Olugbenle Oyegoke, Alexandra Roux, Matthew A. Adeleke, Rajendra Maharaj, Moses Okpeku

**Affiliations:** 1Discipline of Genetics, School of Life Sciences, University of KwaZulu-Natal, Westville Campus, Durban 4041, South Africa; akohseg@gmail.com (O.P.A.); ttai_adewumi@yahoo.com (T.S.A.); leahmaharaj@gmail.com (L.M.); brokunle2002@gmail.com (O.O.O.); alextroux@gmail.com (A.R.); adelekem@ukzn.ac.za (M.A.A.); 2Office of Malaria Research, South African Medical Research Council, Cape Town 7505, South Africa; rmaharaj@mrc.ac.za

**Keywords:** whole genome sequencing, next generation sequencing, malaria, elimination, drug resistance

## Abstract

**Simple Summary:**

Malaria is one of the most severe life-threatening human vector-borne diseases worldwide today, leading to high mortality. Children under the age of five and pregnant women in many developing countries are the most vulnerable groups. However, significant reduction has been achieved globally. This is owed to the advancement in technology. Whole genome sequencing (WGS) is such a high throughput technology, which provides unprecedented relevant information concerning malaria parasite genomes used to study malaria pathology. Here, we present the roles of WGS in malaria elimination. This review also found insufficient availability of WGS within sub-Saharan Africa which bears the highest malaria burden and proposed that if malaria elimination is to be achieved in this region, laboratories should be strategically equipped with WGS machines, where clinical isolates could be received and processed affordably within the region.

**Abstract:**

Malaria elimination remains an important goal that requires the adoption of sophisticated science and management strategies in the era of the COVID-19 pandemic. The advent of next generation sequencing (NGS) is making whole genome sequencing (WGS) a standard today in the field of life sciences, as PCR genotyping and targeted sequencing provide insufficient information compared to the whole genome. Thus, adapting WGS approaches to malaria parasites is pertinent to studying the epidemiology of the disease, as different regions are at different phases in their malaria elimination agenda. Therefore, this review highlights the applications of WGS in disease management, challenges of WGS in controlling malaria parasites, and in furtherance, provides the roles of WGS in pursuit of malaria reduction and elimination. WGS has invaluable impacts in malaria research and has helped countries to reach elimination phase rapidly by providing required information needed to thwart transmission, pathology, and drug resistance. However, to eliminate malaria in sub-Saharan Africa (SSA), with high malaria transmission, we recommend that WGS machines should be readily available and affordable in the region.

## 1. Introduction

Malaria is an infectious, vector-borne and parasitic disease caused by *Plasmodium* species and transmitted through infected female *Anopheles* mosquitoes [1]. It has been a major global health problem to humans through history and is a leading cause of disease and death across many tropical and subtropical countries [2]. It is one of the deadliest infectious diseases, constituting clinical problems and hampering national socioeconomic development, particularly in low income countries [3]. The infection and proliferation of the parasites reduce the number of red blood cells (RBCs), leading to acute anaemia. Furthermore, *P. falciparum* undergoes sequestration and cytoadherence in endothelia cells which may lead to organ failure and cerebral malaria, culminating in death [1]. Compared with infectious, parasitic diseases such as HIV/AIDS, resistant tuberculosis (TB), measles, Middle East respiratory syndrome (MERS), Ebola, Zika, Methicillin-Resistant *Staphylococcus aureus* (MRSA), and COVID-19 (SARS-CoV-2), all of which inflict morbidity and mortality [4], malaria remains the most clinically dangerous and difficult to control. This is due to the intricate life cycle [5,6,7,8,9,10,11,12,13] of the malaria parasite, which complicates the production of clinically effective vaccines and hampers the efficient control of the disease at the infective stage [14]. Additionally, it is one of the most severe life-threatening human vector-borne diseases today, leading to high mortality among young children under the age of five and pregnant women in many developing countries [4]. It is widely distributed across sub-Saharan Africa, southern Africa, South America, Southeast Asia, India, Central America and Pacific Islands, of which sub-Sharan African countries and India share major malaria incidences [5]. The favourable environmental conditions of insect vectors have sponsored both intra (autochthonous) and inter (imported) transmissions across malaria endemic regions [6,7].

In the last few decades, the global malaria burden has experienced a significant reduction in incidence cases and death [8] through robust vector control management systems and suitable different drug policies adopted in various regions. However, this progress has been brought to a halt due to a resurgence in malaria cases [9] and the spread of some resistance genotypes observed in recent years [10,11,12,13]. Therefore, robust surveillance with sophisticated tools is needed to achieve malaria elimination. These tools should be capable of detecting low parasitaemia in an asymptomatic human host, discriminating between reinfection and recrudescence for proper characterisation of *Plasmodium* population [14]. Presently, malaria diagnosis is largely dependent on rapid diagnostic test (RDT) kits and light microscopy (LM); many countries have incorporated RDTs into their surveillance system which allows for quick and easy diagnosis of malaria infection in the field [15]. Contrastingly, in the recent times, RDT kits have been found to generate false negatives due to the deletion of the *PfHRP* antigen [16] and are also unable to resolve mixed infections in samples [17]. This forces some laboratories to perform further confirmatory microscopy test on samples, though laborious and time consuming. Furthermore, there is insufficient microscopy experts who could accurately identify malaria parasites [18]. Therefore, the era of genomics shifted the paradigm to a sequence-based approach where parasites are detected through molecular assays, and information on population structure, transmission and resistance are detected at a nucleotide level [19]. This knowledge answers some important biological questions that inform decisions leading to malaria elimination in most countries [20]. 

Different genotyping methods have been used to study malaria parasites. For instance, high resolution melting assays were used to study the geographic origin and transmission of imported *P. falciparum* malaria cases [21]. Diversity studies of merozoite surface protein 1 and merozoite surface protein 2 was recently studied using PCR genotyping [22]. Sometimes, genotyping assays are combined to achieve better resolution of malaria parasites [23]. Despite these improvements, there are still limits in their capacity to generate high throughput data to capture the needed information at a whole genome level. For instance, PCR genotyping is not able to detect relevant structural variants useful for downstream applications [24]. The quest for deeper resolution and wider coverage of genomes led to WGS by next generation sequencing (NGS) platforms. NGS technologies produce greater sequence reads per instrument run and at a low cost [24]. Presently, WGS is being widely applied in clinical settings [25,26], coupled with genome-wide variant analysis, it is replacing polymerase chain reaction and previous genotyping tools in discovering medically important variants such as single nucleotide polymorphisms, copy number variants, structural variants etc. [27,28]. 

Global malaria reduction has been made possible through the advent and application of NGS to malaria research. NGS platforms allow millions of sheared fragments of DNA to be sequenced in parallel, generating massive data for further analysis [25,29,30]. The plethora of different NGS platforms has allowed for affordability of cost and accessibility in malaria research, particularly in advanced countries [31]. These NGS platforms increase sequencing capacity by almost 1000 fold [25], making whole genome sequencing available for most investigators [30], contrary to genotyping techniques adopted before the advent of NGS. The major weakness of the genotyping techniques that predated NGS is the limitation in information provided. In contrast, NGS allows for higher resolution and coverage of *Plasmodium* genome, whether as targeted sequencing where the sequence of a particular region of the genome is determined, or the complete determination of an organism’s sequence, called whole genome sequencing (WGS) [32]. The incorporation of WGS to malaria research affords the generation of high-throughput data needed for in-depth probing of malaria pathogens by providing comprehensive genome and transcriptome sequences of *Plasmodium* species [30], which could expedite malaria elimination; as various needful information including evolution of parasites, novel drug resistance loci discovery etc. can be deciphered. Breakthroughs achieved in *Plasmodium* genomics is owed to the increase in high throughput DNA sequencing performed as the costs of WGS continues to decrease [31]. 

The application of WGS to malaria research began in the mid-1990s when there was an emergence and increase in drug resistance. This led to sequencing of the *Plasmodium falciparum* genome to expedite research in order to develop new drug targets and potential vaccine candidates [33]. Since then, WGS has been applied to different aspects of malaria research, as shall be discussed in this review. Therefore, it is essential to provide an in-depth coverage of the various roles WGS has played in the malaria elimination agenda in this genomic era. There is, however, paucity of information on the applications of WGS to malaria research. Therefore, this review provides an overview of strengths and limitations of current WGS platforms, brief health applications of WGS, and a coverage of the different roles WGS plays in pursuit of malaria elimination. We identified gaps in infrastructure, funding and technological expertise, which limit the optimal performance of WGS in low-income regions and therefore suggest possible solutions that would speed up malaria elimination.

## 2. Summary of a Whole Genome Sequence Pipeline

Pipelines or workflows involve a series of steps implicated in executing a raw sequence data file through issuing various command lines by a third party [34]. The bioinformatics pipeline used is important as it determines the speed and accuracy of the research [35]. There are different pipelines developed by different institutions and research laboratories. Here, we summarize the processes common to bioinformatics pipelines (Figure 1) and highlight the challenges limiting pipeline applications.

The resulting data from NGS is usually presented in fastQ format [36]. For reproducible and accurate results, sequences are trimmed to remove “noise”, a process called quality control [37]. The next procedure is the alignment and assembly where the sequence reads of the samples are assigned a location in a known genome by computational algorithms. The frequently used tools for alignment are hash-based and Burrows–Wheeler transform [38,39]. After alignment comes sequence assembly. This is a stage where sequences are merged into large contiguous segments [38]. Two types of assembly exist: De novo assembly and known or reference-based assembly. De novo assembly involves the use of computational algorithms to construct sequence reads to generate novel large overlapping reads known as contigs [40]. Reference assembly aligns each read to an established known genome (reference genome) sequence in order to build a consensus sequence with a minimal identity to the known reference sequence [41,42]. The step which gives meaning to the previous processes in WGS is the variant calling [36]. This stage is crucial to further analyses such as genetic disease associations, detecting mutations resulting in cancer [43], population structure analysis [44,45,46], and evolutionary analysis [47,48,49,50,51]. This helps to locate putative differences which provides information on DNA variants and gives implications and evidence for further biological analyses [35,41]. After variant calling comes genome annotation. Genome annotation involves recognizing genome segments of known and probable open reading frames and comparing the recognized segments to the detected gene sequences from the existing databases [52]. 

However, as useful as bioinformatics pipeline is, many challenges currently limit its usage. The major challenge of the WGS pipeline in malaria research is that huge data generated from WGS requires strong computational expertise and computer machines to extract meaningful information from WGS data [53]. Furthermore, bioinformatics pipelines are highly sophisticated and complex in their statistical and algorithmic approaches. Attempts to resolve these have led to continuous changes in the pipelines, leading to vast different open-source tools that require their own set of software and algorithms. This leaves researchers with different factors to consider in pipeline analysis. Additionally, bioinformatics pipelines are complex to execute; they require intricate command-line instructions to run. This required technical expertise limits its broad usage; thus, there is a need to make pipelines simple and easy-to-use so that clinicians and wet lab researchers can take full advantage of these software tools [54]. Moreover, the process of alignment could be computationally intensive and time consuming, thus requiring high processing and patience. Moreover, it can be difficult to accurately estimate a pipeline’s false-negative rate, as observed in the difficulties in identifying phased variants [36,52].

## 3. An Overview of Applications of Whole Genome Sequencing in Disease Management

Whole genome sequencing has many applications in disease management, ranging from disease surveillance, clinical applications, public health, cancer management etc (Figure 2).

### 3.1. Disease Surveillance

WGS analysis, which makes use of parallel sequencing, enables identification of the source of various unidentified disease aetiologies through screening of many loci for disease causing variants and genomic signatures of infectious pathogens [55]. For instance, the strain discriminative power of WGS enables efficient surveillance of foodborne bacteria pathogens and disease outbreaks, case definitions and affirmation. In metagenomics, WGS analysis was used to discriminate persistent and sporadic *Listeria monocytogenes* in food-related environments [56]. More so, the high resolution of WGS helps to accurately track the source of infection during disease outbreaks [57]. Furthermore, WGS affords a faster all-round in silico microbial typing approach compared to the routine serotyping and molecular subtyping [58,59]. 

### 3.2. Public Health 

#### 3.2.1. Strain Diagnosis in Antimicrobial Resistance

Whole genome sequencing has direct impacts in public health, particularly in antimicrobial resistance [52]. Due to inaccurate identification of microbial strains, antimicrobial resistance (AMR) threatens public health [60]. However, owing to the high throughput WGS, accurate strain diagnostics, AMR profiling, and tracking the sources of recurrent infections and transmission between patients have revealed sufficient information for scientists to proffer solutions to AMR [61]. Furthermore, when screening for *cefotaxime*-resistant *Escherichia coli* (CREC), matrix-assisted mass spectrometry identified one resistance strain, but when subjected to WGS, different antimicrobial resistance strains and virulence genes were identified (Figure 2) [62]. This makes WGS a prospective microbial diagnostic [61,63].

#### 3.2.2. Microbial Diagnosis and Evolutionary Studies

WGS was able to unravel transmission patterns, evolutionary relationships, antimicrobial resistance mechanisms, and resistance vehicles driving the expansion of some resistance phenotypes that resulted in carbapenem resistance in the Philippines [64]. WGS has also helped to achieve an expedited and cheaper susceptibility testing of pathogenic microbes compared to the usual routine workflows in *Mycobacterium tuberculosis* research [26]. It also expedites detection of *Mycobacterium tuberculosis* resistant isolates [65]. It provides a powerful technique for rapid identification of genomic variations and various mutations in a given strain in a single step [66], as exemplified in epidemiological study of *Staphylococcus aureus* in a clinical setting (Figure 2) [67]. Furthermore, it provides insights into the evolution and dynamics of methicillin-resistant *S. aureus* [68], antimicrobial resistance, genetics of pathogenesis, development and spread of lineages, and the population structure of *S. aureus* [69]. In addition to traditional microbial typing, WGS is recommended for further confirmation of strains identification, investigation of additional virulence factors and antimicrobial resistance genotypes [70], because the evolutionary and clonal discrimination provided by WGS analyses permit researchers to link related illnesses to their respective candidate genes, which traditional techniques could have missed [63]. This was corroborated when the efficiency in strain discrimination was compared and WGS gave a superior and accurate discrimination than conventional typing [71]. Additionally, WGS is a propitious analysis that gives precise chemically induced mutations, as evidenced in the genome of *C. elegans* [72]. Moreover, the diagnosis, treatment, surveillance, investigation of outbreaks and evolution of individual strains of human and animal *Mycobacterium tuberculosis* complex (MTBC) lineages may also be traced with whole genome analyses (Figure 2) [73].

## 4. Different Whole Genome Sequencing Platforms

Recently, many companies have developed various machines capable of performing WGS. They include Roche, Illumina, SOLiD, Oxford nanopore, etc. Here, we discuss different platforms by looking at their strengths and weaknesses rather than their known mechanisms of operation. Table 1 comprises different WGS platforms.

### 4.1. Roche 454 Pyrosequencing

Technically, Roche 454 sequencer was the first commercial platform to generate long read sequences [74], thus making it easy to overcome the bottlenecks of library preparation and genome assembly of *P. falciparum* [75]. The Roche/454 (Branford, CT, USA) system’s strength is its longer sequence reads generated within relatively short hours. During the process, the Roche/454 GS FLX can create over 1 million individual sequence reads with read lengths of about 400 bases. The Roche/454 system is most suited for de novo sequencing of novel genomes, where long read length is necessary for de novo genome assembly [76]. The main challenges of the sequencing technique include the high cost of polymerase reagents used during the DNA polymerase enzymatic system and a high mistake rate with repeated nucleotide inclusion [77] during variant calling. Additionally, there are nucleotide insertions and deletions of protein coding regions of the genome but there exist fewer substitution mistakes than insertions or deletions in 454 sequencing [78]. 

In addition, its drawbacks include the limited amount of data generated, which results in a relatively expensive cost, and the difficulty in handling homopolymers [79]. Moreover, due to an estimate of 80.6% (A + T) rich *Plasmodium* genome, there is a limitation in genome-wide coverage as it results to low hybridization efficacy and difficulty in reproducing results. This constitutes a major challenge in using Roche 454 in sequencing nucleotide bias genomes such as malaria parasites’ genomes [74]. Because AC microsatellite variants have much lower coverage than their AT counterpart, the 454-platform may experience difficulty in sequencing AC regions of homopolymers in variants and therefore yield low accuracy and variant detection *Plasmodium* genomes [80,81]. Owing to its long read capacity, 454-platform is used to generate de novo sequences in *Anopheles funectus* [82], and *P. cynomolgi* [83] in transcriptome profiling of pyrethroid resistant mosquitoes from China [75], transcriptome analysis of female *An. albimanus* [84] and analysis of species and genotype diversity in Gabon [85]. Roche technology helps to perform de novo sequencing and deciphers disease in novel vectors. This knowledge has helped in transmission control and devised elimination programs, leading to containment of the disease at the vector level and thereby reducing malaria burden or achieving elimination [86]. 

### 4.2. Illumina (Solexa)

The Illumina sequencer (San Diego, CA, USA) varies from the Roche 454 machine in that it generates a larger output with fewer reagents by using detachable fluorescently-labelled chain-terminating nucleotides. It could be said that the (A + T)-rich genome which constituted a “reading” problem for Roche 454 sequencers spurred the development of various Illumina platforms, which did not only overcome the (A + T)-rich challenge but also produced higher reads per run, which makes it a more acceptable sequencer for *Plasmodium* genome [87], even though there is an overrepresentation of GC-rich areas and an underrepresentation of AT-rich regions [79]. In addition, the ability to sequence the *Plasmodium* genome at a greater depth and wider coverage within a limited time and at a relatively cheaper cost has made Illumina sequencing the preferable sequencing machine in the world of *Plasmodium* sequencing today [24]. Although the Illumina platform generates high throughput data at reduced costs, its major limitation is the generation of short read sequences [88]. Moreover, another weakness of Illumina sequencer is the production of large data sets (short reads), creating a major algorithm problem in mapping to reference the genome [81]. 

Furthermore, bases lacking a fluorophore can cause leading or lagging dephasing. Chemical cleavage of terminal moieties and florescent dye labels are also susceptible to failure. As a result, Illumina platforms produce substantially shorter reads, with replacements being the most common error. Due to inappropriate incorporation of nucleotides, “dephasing noise”, the base-call error rate rises with read length. In the diversity study of *P. vivax* isolates from Asia, Latin America and East Asia, Illumina sequencing was used for *P. vivax* library construction and to generate about 150-fold coverage [89]. The Illumina sequencing platform was also used in an analysis of *Plasmodium falciparum* isolates from Asia and Oceania, which showed diversity in natural infections [90,91]. Additionally, the Illumina sequencing platform was used in sequencing *P. vivax* isolate from Peru [92]. 

Notably, Illumina sequencing platforms have played pivotal roles in malaria elimination in malaria-free regions due to its availability and affordability in high income countries [93]. The affordability of Illumina platforms makes it the most frequently used sequencing platform in malaria research. This results in a paradigm shift in malaria research as an accurate detection of transmission level [94], tracking parasites’ importation [95], and resistance genes [96] lead to a reduction in or elimination of malaria [87].

### 4.3. Sequencing by Oligonucleotide Ligation and Detection (SOLiD)

Primarily, among others, sequencing by oligonucleotide ligation and detection (SOLiD) is used for DNA methylation analysis of genomes, particularly the *Plasmodium* genome as it offers a cost-effective sequencing platform. The ABI SOLiD platform is the first commercial platform to provide an improved long read sequence with the highest accuracy of roughly 99.99% compared to other sequencing platforms of its age which produced short reads [97]. Because of the ability of the strong reads generated by SOLiD, it is seen to be a prospective sequencing platform for a bisulphite-converted whole genome that could allow for the analysis of DNA methylation across the genome [98]. 

In addition, the SOLiD system reduces homopolymeric sequencing errors (common in 454-Roche Technology) by employing ligation-based sequencing-by-synthesis, allowing most read errors to be discovered and corrected using the constructing two-base encoding method [76]. However, as is commonly observed in nucleic acids, palindromic regions in nucleic acids are either read as pseudo sequences or not read at all by SOLiD sequencers that make use of the sequence-by-ligation mechanism. This fact is further complicated as readable sequences, two-base pairs prior to the palindromes, are read faintly. Coupled with the fact that Solexa and 454 platforms make use of DNA polymerase which is found within the cell, these platforms are more acceptable in malaria research than SOLiD [99]. Moreover, as with other NGS platforms, bioinformatics pipeline of SOLiD is more expensive and more complicated to use [97]. Substitution errors constitute the most common sort of error. The raw error rate is substantial, ranging from about 2% in the 5′ end to about 8% in the 3′ end [76]. These are factors that limit the usage of SOLiD in malaria research despite its accuracy in generating long reads. Moreover, SOLiD platform has opened an alternative path to studying malaria parasites at the epigenetic level. The epigenetic study of malaria parasites proves to be the panacea to malaria elimination as epigenetic drugs are emerging as potent antimalarials [100]. 

### 4.4. BGI Retrovelocity

This is similar to using exact call chemistry in SOLiD sequencing [101]. The BGISEQ-500 (Beijing Genomic Institute, Beijing, China) is comparative in performance to Illumina platforms in sequencing the whole genome and whole transcriptome [102]. A 96.5% to 97% concordance in performance between BGISEQ-500 and HISEQ-4000 (Beijing Genomic Institute, Beijing, China) was reported in whole exome sequencing, showing BGISEQ-500 as a good sequencing platform [103]. This platform, now, is predominantly used in China and a few Asian countries. Compared to Illumina, 454 Roche and SOLiD technologies, BGISEQ-500 is relatively uncommon in sequencing malaria parasites. 

### 4.5. The Ion Torrent

The platform’s success is due to the integration of a chip with millions of complementary metal-oxide semiconductor (CMOS) sensors in its matrix, which allows for the cost-effective and simple compilation of all data [104]. The strengths of this platform include simplicity in operating the instrument, the relatively cheap cost, and the short execution time. Contrastingly, the error rate is quite high and requires more hands-on. In addition, unlike Illumina and 454 Roche technologies, Ion Torrent does not have much acceptance in the market despite its dexterity in adapting several types of chips to different projects. However, this technology has been used in sequencing *Plasmodium* clinical isolates from Kenya to study the parasite population dynamics, transmission and tracking the parasite origin [105], and performed deep sequencing of *Plasmodium falciparum* to estimate allelic diversity in AMA1 antigen in clinical isolates from the Democratic Republic of Congo [106]. In addition, Ion Torrent has been used for sequencing of the merozoite surface protein-1 region to discover genetic hallmarks of *Plasmodium vivax* recurrence [107]. This discovery has made msp1 a potential vaccine candidate under development [44], and, if found effective, would lead to malaria elimination.

### 4.6. Single-Molecule Real-Time (SMRT) Sequencing by Pacific Biosciences

A molecule of DNA polymerase attaches to the bottom of every well through the biotin–streptavidin system in nanostructures [108]. Pacific Bioscience created the third-generation sequencing technology known as single-molecule real-time (SMRT) [109]. SMRT offers luminescence, signal variance over time, which might be useful for forecasting structural changes in the sequence, significant in epigenetic studies such as DNA methylation [110]. SMRT is one of the newest technologies that offers unusually long reads and at a very short time which makes it a promising diagnostic tool of infectious diseases [111]. A major challenge in sequencing the *Plasmodium* genome is the AT-richness at chromosome telomeres. This constitutes a major advantage to parasite survival, as most virulence genes of the parasites are located at the telomeres. However, with the advent of SMRT technology, not only does SMRT generate long read sequences and help in de novo assembly of telomeric AT-rich *Plasmodium* genome, but it also provides information (virulence genes, drug resistance genes and disease transmission) needed for malaria elimination [112]. Some hyper-variable virulence genes such as VAR2CSA, are difficult to sequence and inability to sequence these types of genes will extend the malaria elimination goal. However, the advent of SMRT affords sequencing of these genes from clinical samples to discover and provide information needed for malaria elimination [113]. In addition, the whole genome sequenced *Plasmodium* 3D7 contains fewer gametocytes under in vitro culture conditions, thus, making the intraerythrocytic stage of infection challenging to study. Therefore, an alternative gametocyte-rich strain is needed to be sequenced and annotated. This was achieved in NF54 strain through SMRT sequencing, thus providing the gametocyte-rich strain needed to study the intraerythrocytic stage of the disease, which is essential for malaria elimination [114]. 

### 4.7. Helicos Sequencing by the Genetic Analysis System

Helicos Biosciences has a device called the HeliScopeTM that can process 1.1 Gb each day (http://www.helicosbio.com, accessed on 8 February 2022), allowing billions of molecules to be sequenced in parallel [115]. The Helicos’ true single molecule sequencing technology significantly increases the speed and decreases the cost of sequencing. The Helicos platform is well suited to RNA sequencing (RNA-Seq) as it relies on tag counting or direct RNA-Seq [76]. A major strength of Helicos is the absence of amplification biases such as the underrepresentation of GC-rich or poor regions, however, it stills produces short read lengths [116]. 

### 4.8. Nanopore Sequencing by Oxford Nanopore Technologies (MinION and PromethION)

Using nanopores for DNA and RNA sequencing has been proposed as a solution, because the conductivity of ion currents in the pore changes as the length of a strand of nucleic acid passing through it varies [117]. Unlike Illumina, 454 Roche and SOLiD technologies, Oxford nanopore technology (ONT) offers an “on-field” usage due to its portability. Nanopore sequencing differs from prior technologies in that it directly detects sequences without active DNA synthesis when a lengthy stretch of ssDNA passes through a protein nanopore, which is maintained in an electrically resistant polymer layer [118]. If not for its high errors in sequencing [9], it should be the most frequently used sequencing technology as it is portable, requires little technical expertise and generates long reads at a relatively cheap cost. ONT technology has a better advantage regarding the use of sequencing technology in low income countries as it maintains a simple sample processing step and does not require skilful laboratory technicians [119]. For instance, MinION (Oxford Science Park, Oxford, UK) devices are so portable that they can be run from the USB (Universal Serial Bus) port of personal computers to allow sequencing to be performed instantly on the field of sample collection. The absence of the image analysis stage helps in the quick detection of target DNA screening of pathogens from clinical samples and creates no limit to the length of DNA to be sequenced [120]. Its usage cuts across targeted sequencing, as seen in ultra-deep sequencing of *Kelch13* genes in *Plasmodium* parasites [119], diagnosing *C580Y* mutation of *P. falciparum* [121], and in combination with loop-mediated isothermal amplification in diagnosing human malaria parasites [122]. It was also used to generate improved quality genomes of *An. colluzzi* and *An. arabiensis* [123]. To achieve malaria elimination, early examination of parasites’ genotypes to avoid the spread of drug resistant parasites strains [119], and unravelling the disease dynamics at the vector level are important, and these can only be made possible by a sequencing approach that is accessible, affordable and easy to use, which are all embedded in ONT technology, therefore making the technology a possible solution to malaria elimination in malaria endemic regions across low-income countries. 

Illumina sequencers are presently the predominant sequencing platform adopted by different users as it presents affordable and accessible sequencing services to hospitals, universities, laboratories and research institutions as seen in Table 1. It has a high throughput and can sequence many genomes quickly and at a low cost. Many researchers and scientists rely on and patronize its use because of its cost. Nonetheless, its short read assembly is a disadvantage in its operations [97]. 

In conclusion, the advent of WGS plays a pivotal role in reducing malaria incidence cases as evidenced in regions that recently reached malaria pre-elimination or elimination phase. In China for instance, incorporation of WGS into their elimination strategy helped to reduce malaria cases from 30 million into zero indigenous cases in 2017 [124]. Additionally, integration of sophisticated genomic tools into the malaria surveillance system allowed Sri Lanka to achieve malaria incidence reduction from 264,549 in 1999 to 23 indigenous cases in 2012 [125,126]. However, there is paucity of information on the impact of WGS in reducing malaria cases in low- and middle-income countries. This probably explains the reason for continuous heterogenous transmission experienced in the region.

## 5. Sequencing of *Plasmodium falciparum* 3D7 Genome

The development of WGS has transformed the field of malariology and verified it to be fundamental to identifying specific gene roles and their association with disease conditions [127]. WGS is now an important tool for the surveillance of public health and the molecular epidemiology of infectious diseases such as malaria, and in antimicrobial drug resistance [128]. WGS is being used more frequently to study the biology, cause, distribution and ecology of malaria parasites as it provides a complete picture of its genome [129]. In 1996, a synergy of global effort was initiated to sequence the *Plasmodium falciparum* genome with the anticipation that it could unlock new avenues of malaria research [130]. A breakthrough was observed when the whole genome sequence of the 3D7 strain of *P. falciparum* was completed by Gardner et al. (2002), thus providing new opportunities in malaria studies as whole-genome gene expression profiling methods can now be utilized [131]. The 3D7 strain of *P. falciparum* has a 23-megabase nuclear genome made up of 14 chromosomes, it encodes approximately 5300 genes and has an overall (A + T) composition of 80.6% which increased to approximately 90% in introns and intergenic regions [130]. 

## 6. The Role of Whole Genome Sequencing in Malaria Elimination

For malaria studies, WGS is extensively being used to study the biology, epidemiology, and ecology of *Plasmodium* parasites [129]. WGS has been used in the surveillance of insecticide resistance in mosquitoes [132], characterisation of genetic diversity and population structure in *Plasmodium* species [28,44,133,134], and identification of potential markers of drug resistance in humans [135]. Here, we extensively review how WGS has contributed to malaria elimination (Figure 3).

### 6.1. Parasites Population Structure and Geographic Origin

In order to design control measures and analyse the effectiveness of elimination strategies, genetic surveillance of malaria parasite populations is crucial [136]. A crucial component of successful malaria eradication is the ability to determine the geographic origin of malaria parasites and monitor their migratory patterns in connection to clinically important characteristics such as treatment resistance. In the era where many countries have reduced their malaria global burden, some countries are working hard to just survive, some are at the verge of elimination and others have attained elimination only to have recurrence of the disease repeatedly. Therefore, the study of parasite population structures has become pertinent more than ever in order to understand the dynamics of transmission in the era of elimination [91]. The traditional microsatellite analysis (which is superior to PCR genotyping) generates significant artefacts in amplification and diversity, thus, making population structure studies rigorous. Furthermore, each genotyping platform and laboratory employs different approaches which create difficulty in standardisation. Consequently, microsatellite genotyping is not propitious for malaria elimination [137]. 

Conversely, the advent of WGS analysis has compensated for these limitations; therefore, it is now extensively explored to study the population structure of *P. falciparum*. Therefore, the massive data provided by application of WGS afford malaria researchers the privilege to explore the population structure of malaria parasites [44,138]. This knowledge helped in monitoring the malaria transmission level, local and imported malaria cases, drug resistance genotypes and offered proper drug policy and treatment [139,140], which were pivotal to achieving malaria elimination in many previously malaria endemic regions.

For instance, WGS analysis of *Plasmodium vivax* clinical isolates exhibited significant genetic variations on the China–Myanmar Border (CMB) [141], and major geographical divisions in *Plasmodium falciparum* population structures (Figure 3) [142]. This significant diversity was not limited to *P. falciparum* only, as it was recently discovered that *P. vivax* across 11 countries of the globe was more diverse [140]. Furthermore, analysis of whole genome data discriminatively clustered the *Plasmodium* population into local and imported infections of Chinese residents and Chinese travellers [142], as demonstrated in the independent evolution of parasites from Cambodia and Thailand [143]. Population genetic analysis of WGS data clarified geographical clustering of *P. malariae* [144], *Plasmodium knowlesi*, and *P. vivax* [145]. WGS revealed a sympatric relationship between the largest two *Plasmodium* isolates in the human population. WGS of global patterns of *P. falciparum* between African countries (Kenya, Mali and Burkina Faso) and Asian countries (Thailand, Cambodia and Papua New Guinea) revealed a higher diversity and proposed that *P. falciparum* originated from Africa [91]. 

The population structure study of *P. falciparum* within a country helps to adopt different control strategies in different areas of a country depending on transmission level and parasites origin in a region. Consequently, different strategies could be used within a country depending on transmission level, as revealed by WGS. The study of parasite population structure also allows for intra-specific intervention (ISI), which is the best strategy to reduce transmission, particularly in endemic regions where parasites exhibit differing phenotypes within a region. Due to the parasite’s recombination rate, large effective population size and high rates of gene flow, genome-wide sequence analyses at the population level in African diverse *P. falciparum* populations become crucial because malaria transmission intensity and parasite genetic diversity vary greatly among different parts of Africa due to variations in season and rainfall [46]. The knowledge gap filled by parasites population structure culminates to malaria elimination achieved in some parts of the globe.

### 6.2. Identification of Structural Variants in Plasmodium Species

Identification of the resurgent parasites, a function of parasites variant detection, is crucial to a strategic planning for malaria elimination [146]. Structural variations in genomes occur when there are significant insertions, deletions, duplications and inversions, leading to complex rearrangements in genome bases [147]. Structural variations (SVs) tend to generate greater impacts on determining *Plasmodium* phenotypes than in single-nucleotide polymorphisms (SNPs). SVs are implicated in gene expression variation and systemic autoimmunity; thus, it is important to unravel the genetic underpinnings of structural variation to understand *Plasmodium* phenotypic variations [147]. 

Copy number variations (CNVs), amplifications and deletions of DNA fragments [148,149] are significant genomic structural variations that aid parasites survival phenotypes [150] such as red blood cell invasion, antimalarial resistance, cytoadherence, virulence, transmission and evolution [137]. The CNVs are a thousand more frequent than single nucleotide variations and are a pivotal process that enable parasite genomes to adapt to new environmental stressors [151]. The significant structural changes are pivotal to anti-malarial resistance and host immune evasion that helps the parasites to cause human morbidity and mortality. For example, when *P. falciparum* parasites were exposed to continuous piperaquine pressure in vitro, a copy number variation on chromosome 5 was shown to be linked with piperaquine resistance [152]. Furthermore, utilizing whole genome sequencing, a study of significant structural variation in 2855 clinical *P. falciparum* isolates from 21 malaria-endemic countries, revealed a new duplication of the chloroquine resistance related gene [153]. 

Whole genome sequencing also revealed CNVs to be drivers of metabolic insecticide resistance in *Anopheles gambiae* [132]. In spite of significant contributions of CNVs to malaria transmission, it has received little research attention and, thus, is understudied, as they are more difficult to detect than single-nucleotide polymorphisms [132]. Furthermore, targeted sequencing methods are averagely limited to 100% coverage of the sequence from any targeted region [27]. Through WGS data analysis of *Plasmodium* genomes, mechanisms of CNVs have been unravelled and exploited, leading to a reduction in the parasites fitness [148]. However, amplification in copy number analysis is difficult to track using WGS data of clonal infections and low datasets. Fortunately, different algorithms have been developed to scan short read WGS data for CNVs. This has made WGS an indispensable tool in malaria elimination agenda [137]. 

### 6.3. Measuring Within-Host Genetic Diversity 

Within-host parasite diversity is defined as the number of different genotypes in an infection determined by multiplicity of infection (MOI) [154]. Infections with multiple parasite genotypes is characteristic of malaria endemic regions. This could be resultant of multiple bites with distinct genotypes from different infectious mosquitos (superinfection) or infection of more than one genotype in a single infectious mosquitos’ bite (multi-clonal infection) and may lead to evolution of drug resistance and parasite virulence [154,155]. Competition arising in within-host parasite strains putatively determines the parasites evolution, antimalarial resistance and virulence, which are the parasites survival phenotypes [154,156]. In regions with high malaria transmission, parasite population is expected to have unrelated lineages [157]. 

According to the literature, within-host competition arises when genetically different *P. falciparum* strains co-infect one other, inhibiting the development of other strains. This link might be advantageous since it could aid in the suppression of virulent strains and offer proof that *P. falciparum* genetic variation impacts the severity of specific malaria symptoms [158,159]. The impact of within-host associations has not been well exploited in human malaria due to the insufficient tools for resolving infection complexity in bulk analysis, until the advent of WGS [160]. The analysis of within-host diversity inside a host provides information on the selection forces unique to that host. Within-host studies in young children identified potential amino acid positions contributing to strain-specific immunity. This suggests that effective malaria vaccines should account for allele-specific immunity, which is a requirement for developing effective malaria vaccines [161]. The study of within-host genetic diversity is so crucial in monitoring transmission patterns of malaria parasites [160], as it revealed continual residual malaria transmission in Limpopo Province, South Africa [162]. 

Different molecular assays predated WGS, but these assays could not detect low level minority genotypes of within-infection in large sequence analysis [155]. MOI is an index of within-host diversity measurement determined by PCR genotyping of polymorphic genes, mostly msp1 and msp2. However, this can only be carried out on specific genes of the parasites, leading to insufficient information [27]. However, massively parallel WGS by high throughput NGS technologies has now enabled in-depth genome-wide analysis which releases sufficient information needed to resolve the within-host parasite surviving phenotypes [154]. Thus, application of WGS helps to explore *Plasmodium* within-host diversity that sponsors important *Plasmodium* survival phenotypes, and to delineate in vivo mechanism of drug resistance, and *Plasmodium* plasticity in malaria pathology. An understanding of within-host diversity helps to determine the level of malaria transmission either in the vector, animals, or humans. It also aids in evaluating the effectiveness of previous or current control strategies and to determine if new control strategy is needed. Therefore, the knowledge gap bridged by WGS to understand the roles played by within-host diversity has culminated in malaria elimination in many regions today. 

### 6.4. Detection of Parasites Evolution

The insights that whole genome sequencing has brought to the study of evolution of malaria parasites cannot be overemphasized. The study of molecular evolution could have remained a “hard nut to crack” if a high throughput sequencing such as WGS is still a mirage [27]. Through WGS, many works have been carried out which shed light on the origin and lineages of these malaria parasites (Figure 3) [45,46,138]. In high malaria transmission regions, multiple *P. falciparum* lineages remain a challenge to detect, thus culminating in pathogenesis and spread of drug resistance genotypes [163]. Evolutionary study of malaria parasites is crucial for identification, tracking transmission dynamics, and the evolution of important phenotypes, which has contributed to parasites survival over the ages [47]. 

The process of evolution leads to parasite diversity and estimates of parasite diversity provides room to study infection dynamics, transmission levels, pathogenesis mechanisms, and drug efficacy. This knowledge is relevant to monitoring the disease and evaluation of various deployed interventions [49], even as many countries are working towards elimination of malaria. Prior to WGS, there have been different genotyping mechanisms such as PCR genotyping of polymorphic markers of malaria parasites. An advancement on that was amplicon sequencing, which seemed to give better resolutions to strain-specific parasites as an advantage over PCR. Although these methods still play a significant role in the battle against malaria, being cheaper and easier to carry out, the creation of artefacts inflates overall estimations of genomic diversity and mutation rate, and the production of limited data provided by polymorphic markers [164] limit their capacity to produce enough data needed for a robust analysis required for parasites evolution studies and, thus, prolongs elimination targets in malaria endemic regions. Most interestingly, *Plasmodium* phenotypes such as drug resistance, red blood cell invasion, cytoadherence, and immune evasion are resultant effects of molecular evolution which can only be captured at nucleotide level and requires massively parallel sequencing with high breadth and depth of strain-specific discriminative power, which is lacking in the PCR genotyping but resolved through WGS [27,164]. Therefore, WGS is more robust in studying the evolution of important phenotypes of *Plasmodium* parasites and helps to proffer solutions, which has not only led to a reduction in malaria burden, but has also allowed some countries to achieve their malaria elimination goal [165]. 

In countries such as China that have eliminated malaria, WGS has been incorporated into the malaria surveillance system to monitor parasite dynamics and avoid re-introduction, emergence and spread of resistance phenotypes into the population [5,45]. This provides advantages to extrapolate more information from genetic sequences for evolutionary analysis which PCR genotyping of molecular markers may not be able to provide [164]. Genome-wide analyses of malaria parasites enable the study of important parasites phenotypes such as drug resistance, virulence, within-species and between-species genetic polymorphism and selection, acting both on genes and genomic architecture, resulting in evolution [48,50,166,167,168]. Through WGS analysis, the intra and inter evolutionary relationships between species of *Plasmodium* could be traced [49]. The reference genome of a *vivax*-like *Plasmodium* was generated and discovered to genetically deviate from the human infecting *vivax*, suggesting that many adaptation changes in *Plasmodium* parasites to humans occurred on multiple occasions before their final evolution into human-infecting *vivax* [169]. Moreover, WGS of *P. falciparum* isolates provides data to fully understand population-specific phenotypes such as evolution of antimalarial resistance, detection of potential targets of host immunity and candidate vaccine antigens needed for malaria elimination [49]. 

### 6.5. WGS Helps in Drug Resistance Surveillance

Drug resistance is a term used when parasites express delayed or slow clearance after being exposed to the correct dosage of the same drugs, which previously achieved immediate parasite clearance [169,170,171]. In order words, drug resistance occurs when the parasite strain can tolerate the drug it previously showed sensitivity to, at the same time and dosage [172]. Ideally, resistance is recognised at a low level i.e., drug tolerance, and at low frequency in a carrier. Understanding the genetics underpinning *Plasmodium* adaptations is required to keep resistance at bay [173]. To achieve this, resistance has to be studied at the nucleotide level through a high throughput sequencing method to track the emergence and spread of drug resistant parasite strains in Africa, particularly, sub-Saharan Africa and across the globe [174]. Accurate assessment and tracking of genetic markers of antimalarial resistance across all vulnerable populations will require vast molecular surveillance with advanced technologies and approaches in the era of dynamic malaria epidemiology [175]. 

Prior to WGS, identification of mutations was not only slow, but difficult due to dearth of high throughput tools, thereby leading to poor understanding of genetics underpinning drug resistance in malaria parasites. However, recent advances in WGS have generated high throughput data to investigate the genetic architecture of drug resistance [176]. Recently, WGS analysis discovered common resistance loci in *P. falciparum* genome (pfcrt, dhfr, and pfmdr1), and novel resistance loci which routine array-based approaches did not discover (Figure 3) [139]. Phosphoinositide-dependent protein kinase 1 (PDK1; PVX_091715) and phosphatidylinositol 3 kinase (PI3K; PVX_080480) are some classical examples of novel resistance genes discovered in *P. vivax* by a sequence-based approach which the usual array-based approaches could not have detected [140]. WGS provides better insightful information about the evolution of drug resistance as it enables the genome-wide analysis of parasites [177] and thus provides the information needed to assist in eliminating malaria parasites. For instance, the analysis of WGS data of local *falciparum* population detected resistance genes peculiar to different regions; this knowledge informed change in antimalarial drug policies across these regions, thereby curtailing the emergence and spread of resistance genes [178]. As shall be discussed later, some challenges common to WGS limit its incorporation into routine antimalarial resistance surveillance programs of many countries, particularly in low income regions, thus limiting its usage to research presently [179]. However, if WGS could be incorporated into routine malaria surveillance, it would provide a thorough understanding of pathogen epidemiology, evolution and population dynamics, genetics of resistance strains and loci, thus culminating in knowledge that informs malaria elimination [180]. More importantly, WGS data analysis has given insights into candidate antigens that could serve as potential malaria vaccine candidates and could be a panacea to malaria elimination [45].

### 6.6. Application of WGS in Vaccine Discovery

Vaccination is the most effective strategy to prevent malaria disease. The availability of whole genome sequences, with advancements in high-throughput technologies, has resulted in a new vaccine development paradigm [181]. Several advances have been made in the development of malaria vaccines in the past, but few ground-breaking advances have been achieved. The path to the creation of a malaria vaccine has continued since the 21st century, and its target is entirely dependent on halting the infective blood stage life cycle. This should provide enough protection to the human host when successful [182,183]. Polymorphism of several allele types of *Plasmodium* is a challenge, however, developing a multivalent vaccine is expected to improve the host’s protection. 

Malaria is an ideal paradigm for developing genome-based vaccines because of the rising and growing concerns about malaria in SSA. Immune evasion and pathogenesis of malaria may now be investigated at the molecular level thanks to genomics and, more recently, next generation sequencing. WGS provides a basis for genome-wide screening for antigen identification to locate alleles suited as potential vaccine candidates. For instance, genome wide analysis of *P. falciparum* revealed merozoite surface proteins (msp1, msp4, msp7, msp9), rhoptry neck protein 2, ron2, duffy binding-like merozoite surface proteins (mspdbl1, mspdbl2), rhoptry associated adhesins, serine repeat antigens 6, sera6, and rhomboid proteases and rom4 as invasion related genes [45], out of which msp1, msp7, mspdbl1, mspdbl2, and sera6 were identified as prospective malaria multivalent vaccine candidates [182,184,185], while liver stage antigen 3, lsa3, apical membrane antigen 1, ama1, erythrocytes binding antigen 175, eba175, or circumsporozoite protein, csp are currently being explored as vaccine candidates [182,185]. 

WGS data provide a template for vaccine allele prediction, which could curtail malaria transmission and elimination. It was almost impossible to discover all prospective candidate antigens prior to WGS [182]. Therefore, the most important role played by WGS is the genome-wide scan of all putative vaccine candidates in malaria parasites [45]. This will expedite effective vaccine discovery, which, if developed, will gallantly achieve malaria elimination in endemic regions.

### 6.7. WGS Deciphers Malaria Pathogens in Mosquitoes

The emergence of novel resistance alleles in malaria vectors is a harbinger to malaria control as it increases vectors’ plasticity that sponsors the persistent transmission experienced in areas where vectoral control is not properly executed. The role played by parasite vectors in malaria pathology and transmission cannot be underestimated. The vectors of malaria are crucial to malaria plasticity, thereby making the study of the vector as important as studying the disease in humans. Therefore, for malaria to be eliminated within a region, substantial efforts must be directed toward unravelling the disease at the vector level. The susceptibility of mosquitos to malaria infection is species-specific, and the strength of vectors’ innate immunity is a function of its microbiota. This innate immunity explains the biological differences in vector susceptibility to malaria infection [186] and their parasite transmission capacity. For instance, as revealed by Roche 454 WGS data, the diversity of gut microbiome in mosquitoes varies across the different stages of mosquitoes’ development [187]. This diversity in gut microbiota determines whether a mosquito vector would be vulnerable to malaria infection or not, and this could also, in addition, probably sponsor the insecticide resistance witnessed in the mosquito population recently across SSA [188]. 

Concerning insecticide resistance, Lucas and colleagues found copy number variations (CNVs) to be associated with insecticide resistance through WGS analysis of infective malaria vectors. They identified 28 genes with CNVs that have been positively selected for, and fixed in the mosquito population [132]. The positive selection of these genes in mosquitos’ population has sponsored the emergence, transmission and spread of resistance alleles in humans, posing a threat to malaria elimination. Until recently, most malaria control strategies focused on the infective stage of the parasites in human blood. This was primarily carried out by administering drugs that target and clear the merozoite stage of the parasite in the infected red blood cells (iRBCs). However, it was observed that after the parasites were completely cleared from an infected person, he or she could still succumb to another infection. This led to incorporating the entomological approach to elimination strategies of malaria. Countries such as China [189,190] and Sri Lanka [191,192] have successfully achieved malaria elimination recently by incorporating this WGS approach into their malaria elimination programs [190,191]. 

Moreover, the more the resistance alleles, the likely they are to reach fixation in mosquitoes’ population, which culminates in the transmission and spread of resistance phenotypes in humans. When this occurs, it aids the recurrence of malaria disease, irrespective of the efficiency of antimalarials administered, thereby spurring antimalarial resistance. This phenomenon was substantiated in studies where malaria vectors were resistant to pyrethroids in insecticide-treated nets [193,194,195,196,197,198,199]. Compared to extant genomic tools, WGS has facilitated a better understanding of biological factors that could lead to vector population suppression. WGS has proven to be an efficient tool in highlighting the genetic components involved in the local adaptation of mosquitos to insecticides. This knowledge has helped to inform decisions that have led to mosquito population suppression in Malaysia [200]. When resistance to insecticides (deltamethrin K-Othrine) was experienced in a mosquito population, an alternative method using a novel formulation of deltamethrin K-Othrine (PolyZone) was adopted as outdoor residual spray, leading to larva and vector population reduction in Malaysia [201]. Although still in debate, the concept of gene drive could have great potential in reducing mosquito populations, as self-destructing mosquitoes were engineered, which reduced mosquito population when experimented [202].

This knowledge gap bridged by WGS studies has demystified parasite dynamics in vectors, and adapted to controlling the disease at the vector level [203], therefore, contributing to the significant reduction in malaria burden recorded recently. This achievement was made possible through the generation of high throughput data from WGS of malaria parasites [123,204,205]. 

## 7. Challenges of *Plasmodium* Whole Genome Sequence

When carrying out direct WGS of *Plasmodium* species, the little amount of parasite DNA in a clinical blood sample relative to human DNA is a major challenge [206,207]. Due to the indiscriminate DNA fragmentation in WGS, all organisms’ DNAs in an isolate would be sequenced. Thus, owing to the abundance of human DNA, the sequence coverage of *Plasmodium* genome would be greatly reduced. This, therefore, creates difficulty in reducing human DNA contamination from clinical malaria samples, even in high parasitaemia infections [208]. For example, *Plasmodium* parasitaemia in clinical samples could be as small as 9.2% [209], thus limiting parasite DNA coverage. As discussed below, strong bioinformatics expertise, high performance computer machines, stable electricity and finance constitute limitations (particularly in resource-limited settings) in mining *Plasmodium* genome for downstream applications [210,211]. For instance, many of the pipeline’s complex steps necessitate large amounts of RAM, which many laptops and desktop computers lack [212]. Furthermore, despite advancements in WGS of *P. falciparum*, there are still knowledge gaps to be addressed. For instance, several regions of the genome possess short tandem repeats (STRs), and an unusual abundance of other low complex sequences in both coding and non-coding regions which could influence the phenotype significantly by regulating gene expression [213]. 

Furthermore, despite the great sensitivity of whole genome sequencing, it is challenging to discriminate relapse from reinfection of similar *P. vivax* genomes in areas with high malaria endemicity. Additionally, without genotyping all hypnozoites in the liver, it is almost impossible to differentiate between a heterologous relapse and reinfection [214]. Generally, malaria studies are difficult to conduct because of the technical challenges associated with acquiring clinical samples from the field [215], as well as some species (*Plasmodium vivax*) that present low parasite densities [216]. At the onset of *vivax* malaria for instance, parasite density could range between 0.2 and 1.6/microliters [217]. It is then apparent that samples that are collected must be fully exploited to obtain as much meaningful data as possible. In addition, database curation and maintenances are heavily dependent on bioinformatics expertise and this could be a limitation, especially for the low income countries where there are limited trained personnel. This also might be responsible for low level understanding and interpretation of WGS data for inclusion in regular malaria elimination programs, particularly in malaria endemic regions [218].

## 8. Future Perspectives

WGS is rapidly becoming the new gold standard in scientific research and gradually becoming a clinical diagnostic tool for infectious diseases, particularly in advanced countries. WGS enhances detection, diagnosis, and management of infectious parasitic pathogen outbreaks, leading to containment of many infectious diseases and, thus, improves public health. Such diseases include Tuberculosis [219], Pneumonia [220], Gonorrhoea [221], Measles [222,223], Ebola [224], Cholera [225], COVID-19 [226] and Malaria [227]. The impact of high throughput sequencing is radically reducing the global malaria burden and expediting malaria elimination [227]. The use of WGS means that every gene in the genome can be analysed simultaneously, generating an abundance of data with fewer resources within a short time frame. Small SNPs would be identified as well as larger structural changes such as translocations. The implication is that a person can be screened for genetic traits related to multiple single genetic disorders (cystic fibrosis and sickle cell anaemia) all at once. Furthermore, reanalysis of data becomes easier once the entire genome is available for study, likely not requiring further sequencing to be run. 

Despite the immense applications of this WGS to malaria, its usage is still predominant in the advanced countries where these machines are manufactured, and malaria has majorly been eliminated. Owing to many factors that cut across infrastructure, finance, technical expertise etc., in low-income sub-Saharan (SSA) countries where malaria is prevalent, WGS machines are not easily accessible. Concerning infrastructure; power, strong computer devices, state of the laboratories, are some of the challenges confronting WGS analysis. Talking of finance, even though there is gradual drop in cost of performing WGS, most malaria researchers within low income regions still find it difficult to afford [228]. Thus, sending samples for sequencing oversees and receiving the results electronically becomes unavoidable. This factor leads to the third problem of insufficient technical expertise across malaria endemic regions. 

Generally, novel computing technology and skill acquisition in the research sector has increased, deciphering the meaning of sequencing outputs accurately. Clinically, WGS is gradually being used for diagnostics in the developed world, but its use is still limited as it requires appropriate and correct analysis of sequencing results. This use may be benched until the level of analysis needed can be attained to prevent unnecessary stress to patients and the risk of providing inaccurate interpretation of data generated. Notably, the scientific research sector has the most to gain from WGS as it involves critical analysis of results and builds towards wider applications of the method. While researchers work to improve the use of WGS clinically and to ensure easy and correct interpretation of results, strides are being made towards optimizing the use of WGS in day-to-day laboratories that run different kinds of analysis such as forensics, disease screening and evolutionary science. There is still an abundance of information to be exploited from WGS which can lead to improved vector control and malaria elimination. Therefore, more relevant information should be synthesised from data generated from WGS to achieve malaria elimination globally. 

To speedily achieve malaria elimination, laboratories should be strategically equipped with WGS machines, particularly, Illumina platforms, where clinical isolates could be received and processed affordably within malaria endemic countries. The WHO and relevant international bodies should ensure human capacity is built within this region so that the people would self-reliantly apply the technical expertise coupled with the socio-cultural knowledge of the people to tackle malaria transmission and pathology. Ultimately, there should be significant allocation of funding to secure sufficient high throughput machines, as this will improve capacity development in low-income countries. Furthermore, the intricate nature of WGS should be demystified to allow for easy usage by non-computer experts. Demystification of protocols and reduction in costs would help incorporate WGS into routine malaria surveillance. Therefore, bioinformatics algorithms should be simplified, and analytical tools developed into web-based tools allowing non computer expert researchers to visualize, analyse and probe WGS data sets to mine information that would lead to the elimination of malaria [182]. All these measures would improve WGS and, thus, expedite the pursuit of the malaria elimination agenda.

## 9. Conclusions

The use of WGS in malaria research cuts across human and vector hosts. It is gradually becoming a gold standard for diagnosis as it reveals important mutations, particularly, drug resistance mutations. WGS will open prospective opportunities to conduct genetic testing to discover genetic variants responsible for individuals’ drug metabolism, because drug metabolism is genetic and, thus, varies in individuals. This affects the choice of dosage administered to patients, thus culminating in personalized medicine. This is responsible for drug efficacy and safety, and thereby reduces resistance to antimalarials. Traditional single-gene sequencing produces simplistic results with a one-dimensional view of the genetic state. By analysing the entire genome, a large amount of data is generated that can be applied to the study of treatment of multiple diseases. Even though many manipulations are required to make sense of WGS data, WGS analysis is needed as it contributes to a better understanding of the cellular and molecular mechanisms of infection and immunity, which will aid in the development of new malaria control strategies, such as novel antimalarial and vaccines, as well as the improvement of diagnostics and effective vector control techniques [45]. Generally, WGS provides a depth of information that was not previously found in extant tools. In terms of malarial vaccine design, WGS is providing insights into vaccine candidate genes of interest. While there is still room and purpose for the existing tools, WGS is extremely valuable, and its impacts have demystified parasite transmission dynamics, population structure, origin, antimalarial resistance, allele vaccine predictions and immune responses, resulting in global malaria reduction and elimination in some regions. Contrastingly, technical expertise, finance, sophisticated infrastructure etc. has limited its incorporation into routine malaria surveillance.

## Figures and Tables

**Figure 1 biology-11-00587-f001:**
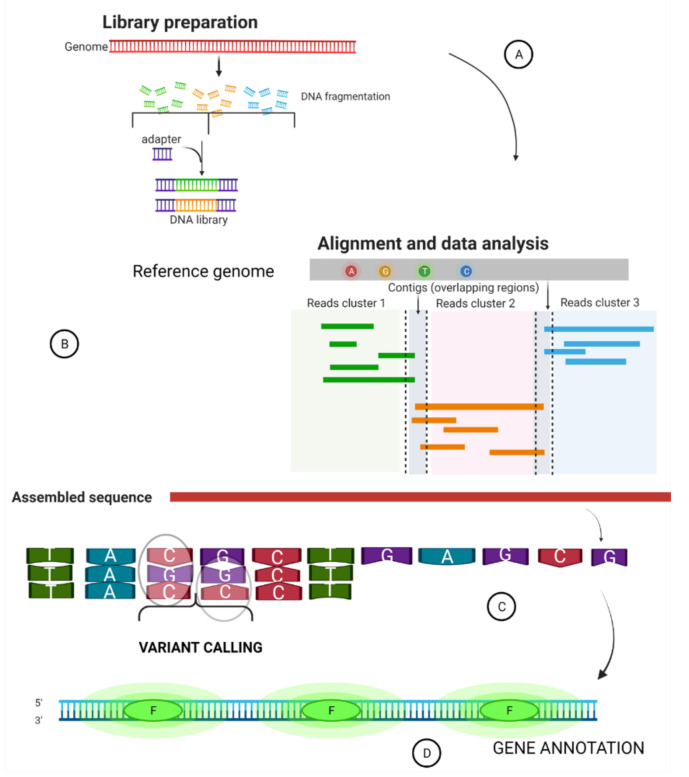
Diagram of a whole genome sequence pipeline. (**A**) Shows process of library preparation. (**B**) Speaks to mapping nucleotide sequence to a reference genome. (**C**) shows detection and calling of variants (SNP). (**D**) Gene annotation: identification of functional elements across the genome. The letters represent functional proteins.

**Figure 2 biology-11-00587-f002:**
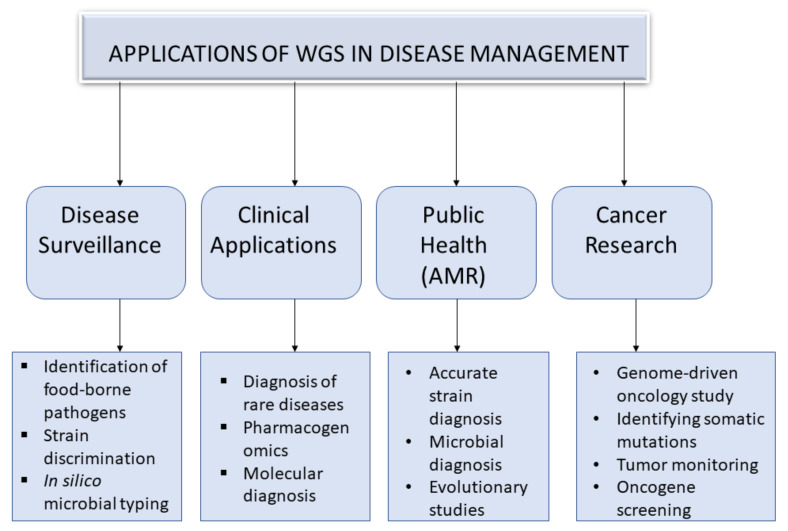
A diagram representing overview of general WGS applications. AMR: *Antimicrobial Resistance*.

**Figure 3 biology-11-00587-f003:**
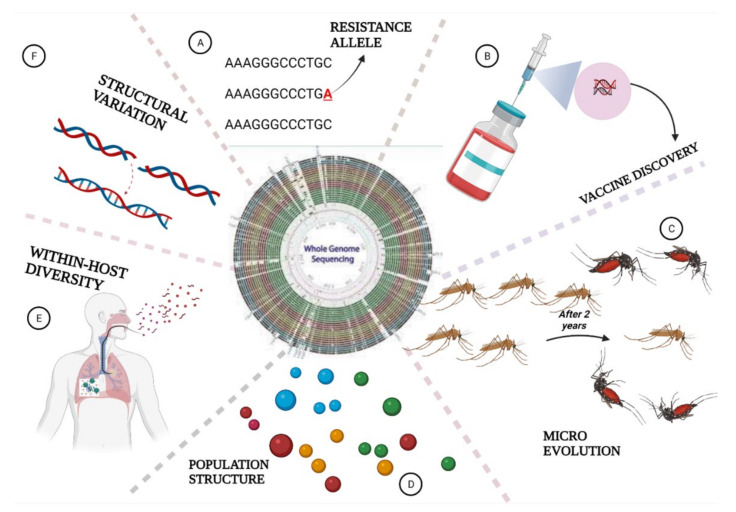
Different roles of WGS in malaria research. (**A**) shows resistance allele (SNP) conferring resistance to antimalarial. (**B**) shows discovery of vaccine candidate genes which may lead to vaccine discovery. (**C**) depicts microevolution from wild to resistant mosquitoes’ population. (**D**) depicts parasites population structure. (**E**) speaks to diverse mosquito genotypes within an infection. (**F**) shows variation in size of DNA. A part of the DNA has been deleted.

**Table 1 biology-11-00587-t001:** Characteristics of different whole genome sequencing platforms.

WGS Platforms	Pros	Cons	Reads
Solexa/Illumina	High precision Analysis with a high throughput The most widely available technology	Read length is rather short. Machines are expensive Could be time consuming	3 G
ABI SOLiD	High accuracy High-throughput analysis	Short read length High cost of machine Time consuming	1.2–1.4 G
Roche 454	Long read length High accuracy Short run time	Expensive machine Moderately low throughput analysis	1 M
Ion Torrent Technologies	Moderate to long read length High throughput analysis High accuracy Relatively cheap when compared to others Shorter run time	No appropriate products	6 × 10^7^
Oxford Nanopore	High-throughput analysis Short processing time	Less accuracy when compared to others Short read length	6 × 10^4^
BGI Retrovolocity	Long reads Short time	Prone to error	1 × 10^9^
Heliscope/Helicos			7 × 10^9^
Pac Bio SMRT	High-throughput analysis Long read length	Prone to error High cost of machine	1 × 10^6^

ABI SOLiD: Applied Biosystem sequencing by oligonucleotide ligation and detection, BGI: Beijing Genomic Institute, Pac Bio SMRT: Pacific Biosciences Single-molecule Real-time, SMS: Single Molecule Sequencing.

## Data Availability

Not applicable.

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
