# Peer review of "Whole Genome Sequencing Contributions and Challenges in Disease Reduction Focused on Malaria"

_biology, 2022, doi:10.3390/biology11040587_

Round 1
Reviewer 1 Report
Globally, malaria perpetuates an enormous disease burden, resulting in over 241 million cases and ~650K deaths, primarily in Sub Saharan Africa as per WHO reports in 2020. With the issues of rapid insecticide resistance cropping up in field and traditional insecticide treated bed nets (ITNs) failing to provide long term relief, novel avenues of disease understanding, and management are urgently required. Akoniyon et. al. provide a comprehensive explanation of Whole Genome Sequencing (WGS) techniques and approaches with a thorough review of different techniques currently in use and their potential application in combating malaria. Although the manuscript is comprehensive, I do have some suggestions -
- While most of the manuscript deals with application of WGS on Plasmodium detection, section 3.2.1, 3.2.2 and 3.4 (Cancer management) distract the reader from the overall goal of the review. In my opinion, these sections should be removed, and the segment ‘Clinical Applications’ should be re-addressed focusing on malaria or other vector borne diseases. The diagram depicting Applications of WGS suffices to put the point across.
- I would like to see more data on the described challenges rather than a simple statement citing the obvious. It would be helpful to see what percentage of malaria cases were reduced or alleviated with the advent of limited WGS application in the region.
- Section 6.7 addresses vector genome plasticity and the potential use of WGS, but no review of potential application/contribution of WGS in driving population suppression or population crash has been described.
- The English language used in the manuscript is riddled with grammatical errors. I have outlined few of the errors, but these run throughout the manuscript. I would highly suggest having the manuscript proofread by a native English speaker.
Minor changes-
Line 131- “A bioinformatics pipeline” should be changes to “The bioinformatics..”
Line 185- “WGS analyses was ..” to be changed to “WGS analysis was”
Line 198- “Also, the aetiology of rare diseases was an problem [73], and rare diseases should better be early be screened for, high throughput techniques, in order to manage and ensure better health of a child.” Revise to form grammatically correct sentence.
Line 203- “candidate genes, thereby, leads to enhanced paediatric diagnosis of rare diseases.” Change to “..candidate genes, thereby leading to..”
Line 507-533- None of the scientific names (P. falciparum etc.) have been italicized.
Author Response
Point-by-point response to reviewer’s comments
The authors appreciate reviewers’ comment and contribution to making our manuscript better. Below is a point-by-point response to reviewers’ comments.
Response to reviewer 1
Comment 1: While most of the manuscript deals with application of WGS on Plasmodium detection, section 3.2.1, 3.2.2 and 3.4 (Cancer management) distract the reader from the overall goal of the review. In my opinion, these sections should be removed, and the segment ‘Clinical Applications’ should be re-addressed focusing on malaria or other vector borne diseases. The diagram depicting Applications of WGS suffices to put the point across.
Response: The authors agreed completely with the reviewer that these sections distract the reader from overall goal of the review, thus sections 3.2, 3.21, 3.2.2 and 3.4 have been expunged as suggested.
Comment 2: I would like to see more data on the described challenges rather than a simple statement citing the obvious. It would be helpful to see what percentage of malaria cases were reduced or alleviated with the advent of limited WGS application in the region.
Response: Data on described challenges have been added and percentage of malaria cases reduced with advent of WGS application was added as suggested.
Comment 3: Section 6.7 addresses vector genome plasticity and the potential use of WGS, but no review of potential application/contribution of WGS in driving population suppression or population crash has been described.
Response: Review on contribution of WGS in driving population suppression in mosquitos was added as suggested.
Comment 4: The English language used in the manuscript is riddled with grammatical errors. I have outlined few of the errors, but these run throughout the manuscript. I would highly suggest having the manuscript proofread by a native English speaker.
Response: The issue of grammatical errors has been fixed as suggested.
Minor changes
Comment 5: Line 131 “A bioinformatics pipeline” should be changes to “The bioinformatics..”
Response: “A bioinformatics” has been replaced with “The bioinformatics”.
Comment 6: Line 185- “WGS analyses was ..” to be changed to “WGS analysis was”
Response: “WGS analyses was” has been replaced with “WGS analysis”.
Comment 7: Line 198- “Also, the aetiology of rare diseases was an problem [73], and rare diseases should better be early be screened for, high throughput techniques, in order to manage and ensure better health of a child.” Revise to form grammatically correct sentence.
Response: This statement was part of the sections the authors removed as suggested by the reviewer.
Comment 8: Line 203- “candidate genes, thereby, leads to enhanced paediatric diagnosis of rare diseases.” Change to “..candidate genes, thereby leading to..”
Response: This statement was part of the sections the authors removed as suggested by the reviewer.
Comment 9: Line 507-533- None of the scientific names (P. falciparum etc.) have been italicized.
Response: The scientific names have been italicized.
Reviewer 2 Report
This manuscript is well written, very interesting and cover a lot in the field.
Howewer there is a major issue at first read for a review, it does not explain the selection of articles inside the review.
Please refer to the guidelines to review in biology publication :
https://www.mdpi.com/journal/biology/instructions#preparation
"Review manuscripts should comprise the front matter, literature review sections and the back matter. The template file can also be used to prepare the front and back matter of your review manuscript. It is not necessary to follow the remaining structure. Structured reviews and meta-analyses should use the same structure as research articles and ensure they conform to the PRISMA guidelines."
In a few words, the diagram should be added for the selection of papers, including the keywords used for research and the different supports (NCBI, google scholar...)
Direct links to prisma diagram :
http://prisma-statement.org/PRISMAStatement/FlowDiagram.aspx
prisma checklist :
http://prisma-statement.org/PRISMAStatement/Checklist.aspx
PRISMA is important because it shows that all sources have been used and explain how some articles could be put aside if they don't match the keywords used by authors.
Minor comments :
Line 50 : "its" should be "the". The sentence goes from line 46 to line 52. It is a bit too long, reader can be lost because of that.
Sentence line 74-76 should be rewritten in comprehensive English.
Author Response
Point-by-point response to reviewers’ comments
The authors appreciate reviewers comment and contribution to making our manuscript better. Below is a point-by-point response to reviewers’ comments.
Response to reviewer 2
Comment: However, there is a major issue at first read for a review, it does not explain the selection of articles inside the review.
Response: After reading biology requirements for review articles again, the authors confirm that selection of articles is not applicable for review articles but is a requirement for structured/systematic review and meta-analysis.
Comment: "Review manuscripts should comprise the front matter, literature review sections and the back matter. The template file can also be used to prepare the front and back matter of your review manuscript. It is not necessary to follow the remaining structure. Structured reviews and meta-analyses should use the same structure as research articles and ensure they conform to the PRISMA guidelines."
Response: The authors confirm that our review article followed the template for review article as directed in instruction to authors. (https://www.mdpi.com/about/article_types)
Comments: In a few words, the diagram should be added for the selection of papers, including the keywords used for research and the different supports (NCBI, google scholar...)
Direct links to prisma diagram :
http://prisma-statement.org/PRISMAStatement/FlowDiagram.aspx
prisma checklist :
http://prisma-statement.org/PRISMAStatement/Checklist.aspx
PRISMA is important because it shows that all sources have been used and explain how some articles could be put aside if they don't match the keywords used by authors.
Response: The authors agreed that these requirements are for systematic reviews and meta-analysis, thus not applicable to our review. (https://www.mdpi.com/about/article_types)
Comment: Line 50: "its" should be "the".
Response: “its” has been replaced with “the”
Comment: The sentence goes from line 46 to line 52. It is a bit too long, reader can be lost because of that.
Response: The sentence has been split, to make more meaning and avoid confusion.
Comment: Sentence line 74-76 should be rewritten in comprehensive English
Response: The sentence has been recrafted to reflect its meaning.
Reviewer 3 Report
This review by Akoniyon et al. provides a clear and exhaustive account on the use of a whole genome sequencing (WGS) approach for genetic surveillance of malaria parasites and on its contributions and challenges towards the global goal of malaria containment and eradication.
The review is well conceived and well written. Moreover, it provides a comprehensive account of the literature as it touches upon all the different aspects of the WGS approach versus the more traditional but less informative PCR genotyping method. As such, I think that it deserves to be published in Biology. However, I believe that some minor amendments should be made to the text of the manuscript prior to its publication:
- The use of italics for the name of species is not consistent throughout the manuscript. Please make sure to introduce it where it is not in use.
- COVID’19 should be COVID-19 throughout the manuscript.
- Line 46-53: this phrase is too long. For the sake of clarity, it should be split and partially revised. For instance, the phrase could read:
Compared with infectious, parasitic diseases like HIV/AIDS, resistant tuberculosis (TB), measles, Middle East respiratory syndrome (MERS), Ebola, Zika, Methicillin-Resistant Staphylococcus aureus (MRSA), and, COVID’19 (SARS-CoV-2), all of which inflict morbidity and mortality [4], malaria remains the most clinically dangerous and difficult to control. This is due to the intricate life cycle [5–13] of the malaria parasite, which complicates the production of clinically effective vaccines and hampers an efficient control of the disease at the infective stage [14].
- Line 105-108: the use of semicolons in this phrase is incorrect. Please revise the entire phrase.
- Line 216-217: This sentence should be rephrased as it is grammatically incorrect. Perhaps, the phrase could read:
The versatility of WGS is not new. Indeed, it was already praised as early as in 2014, when the MedSeq project was published [81].
- In the header of Table 1, please replace Merits and Demerits with Pros and Cons
- Line 319-320: Please remove the comma after “though”, add “the” before “generation” and correct the inconsistent use of singular and plural. Perhaps the phrase could read:
Even though the Illumina platform generates high throughput data at reduced costs, its major limitation is the generation of short read sequences [120].
Author Response
Response to Reviewer 3
Comment: The use of italics for the name of species is not consistent throughout the manuscript. Please make sure to introduce it where it is not in use.
Response: The authors appreciate the observation of the reviewer and species names have been italicized throughout the article.
Comment: COVID’19 should be COVID-19 throughout the manuscript.
Response: COVID’19 is now replaced with COVID-19
Comment: Line 46-53: this phrase is too long. For the sake of clarity, it should be split and partially revised.
Response: The authors split and revised the sentence for better clarity. The section now reads thus: “Compared with infectious, parasitic diseases like HIV/AIDS, resistant tuberculosis (TB), measles, Middle East respiratory syndrome (MERS), Ebola, Zika, Methicillin-Resistant Staphylococcus aureus (MRSA), and, COVID’19 (SARS-CoV-2), all of which inflict morbidity and mortality [4], malaria remains the most clinically dangerous and difficult to control. This is due to the intricate life cycle [5–13] of the malaria parasite, which complicates the production of clinically effective vaccines and hampers an efficient control of the disease at the infective stage” as suggested.
Comment: Line 105-108: the use of semicolons in this phrase is incorrect. Please revise the entire phrase.
Response: The sentence has been revised accordingly.
Comment: Line 216-217: This sentence should be rephrased as it is grammatically incorrect.
Response: The statement has been replaced with “The versatility of WGS is not new. Indeed, it was already praised as early as in 2014, when the MedSeq project was published”.
Comment: In the header of Table 1, please replace Merits and Demerits with Pros and Cons
Response: The Merits and Demerits in Table 1 header have been replaced with Pros and Cons.
Comment: Line 319-320: Please remove the comma after “though”, add “the” before “generation” and correct the inconsistent use of singular and plural.
Response: The statement has been replaced with “Although the Illumina platform generates high throughput data at reduced costs, its major limitation is the generation of short read sequences.” as suggested.